# Exploring General Practitioners’ Knowledge, Attitudes, and Practices towards E-Cigarette Use/Vaping in Children and Adolescents: A Pilot Cross-Sectional Study in Sydney

**DOI:** 10.3390/ijerph21091215

**Published:** 2024-09-16

**Authors:** Rajiv Singh, Michael Burke, Susan Towns, Muhammad Aziz Rahman, Renee Bittoun, Smita Shah, Anthony Liu, Habib Bhurawala

**Affiliations:** 1Department of Paediatrics, University of Toronto, Toronto, ON M5G 1X8, Canada; 2Department of General Practice at the University of Western Sydney, Sydney, NSW 2560, Australia; 3Department of Adolescent Medicine Unit, The Children’s Hospital at Westmead, Sydney, NSW 2145, Australia; 4Faculty of Medicine and Health, The University of Sydney, Sydney, NSW 2050, Australiaanthony.liu@health.nsw.gov.au (A.L.); 5Institute of Health and Wellbeing, Federation University Australia, Berwick, VIC 3806, Australia; 6Lifestyle Medicine, Nicotine Addiction Unit, Avondale University, Sydney, NSW 2265, Australia, Australia; 7Faculty of Medicine, The University of Notre Dame, Sydney, NSW 2007, Australia; 8Department of Paediatrics, Nepean Hospital, Kingswood, NSW 2747, Australia

**Keywords:** e-cigarette, vaping, general practitioners, children, adolescents

## Abstract

(1) Background: The increasing use of e-cigarettes/vaping in children and adolescents has been recognised as a global health concern. We aim to explore the Knowledge, Attitude, and Practice of General Practitioners (GPs) in Sydney regarding the use of e-cigarettes in children and adolescents and identify the barriers to addressing this issue. (2) Methods: This pilot study was a cross-sectional study conducted using an electronic questionnaire with a Likert scale and free-text responses. (3) Results: Fifty-three GPs participated in the study (male = 24 and female = 29) with a mean age of 50 ± 5.5 years. There was strong agreement (mean 4.5) about respiratory adverse effects and addictive potential. However, there was less awareness of cardiac side effects and the occurrence of burns. There is a lack of conversation about e-cigarettes in GP practice and a deficit of confidence in GPs regarding managing e-cigarette use in children and adolescents. (4) Conclusions: Our pilot study has shown that GPs are somewhat knowledgeable about the potential adverse effects of the use of e-cigarettes in children and adolescents, though there is a lack of knowledge of the complete spectrum of adverse effects and more importantly, there is a paucity of a structured approach to discuss the use of e-cigarettes with children and adolescents, and there is a low level of confidence in addressing these issues. There is a need for educational interventions for GPs to increase awareness of the potential adverse effects of using e-cigarettes and build confidence in providing management to children and adolescents regarding the use of e-cigarettes.

## 1. Introduction

E-cigarettes are electronic devices powered by a lithium-ion battery designed to vaporise a liquid stored in cartridges for inhalation (commonly referred to as vaping). Most cartridges contain propylene glycol (PG), glycerol, and water but can also include nicotine, tetrahydrocannabinol (THC), vegetable glycerin (VG), and flavourings [1,2]. Since the introduction of e-cigarettes in 2004, there has been a significant increase in their use by youth [1,2,3], increasing by over 19% between 2011 and 2018 in the United States of America (USA) [4]. Comparable findings have been reported in Canada and the UK [5]. In Australia, Guerin reported that in 2018, around 14% of 12 to 17-year-olds had tried an e-cigarette at least once [6]. Since then, a sharp increase has been noted in the use of e-cigarettes among young people between 2019 and 2022–2023, where the use of e-cigarettes increased from 1.8% to 9.7% among people aged 14–17, making this cohort the third highest in the proportion of e-cigarette use [7].

Concomitantly, there has been an increase in its reporting of adverse events and side effects related to the use of e-cigarettes/vaping. In 2019, more than 1000 new E-cigarette, or Vaping, product use-Associated Lung Injury (EVALI) cases were reported in the USA [8,9]. Sund et al. reported few international cases of EVALI outside the USA, though they raised concerns about possible underreporting of such cases [10]. In addition to the respiratory adverse effects, studies have demonstrated an increased risk for myocardial infarction [11], an increase in arterial stiffness and conducting airway obstruction [12], impairment of vascular endothelial function [13], and burns with the use of e-cigarettes [14]. It is thus clear that the use of e-cigarettes can potentially increase the risk of a magnitude of health problems.

The use of e-cigarettes in youth has been recognised as a global health concern [15]. In July 2024, the Australian Institute of Health and Welfare (AIHW) stated that though the smoking of combustible tobacco has decreased (down from 12.2% in 2016 to 8.3% in 2023), the use of e-cigarettes has significantly increased (from 0.5% in 2016 to 3.5% in 2023). In 2019, only 9.6% of people aged 14 to 17 in Australia had ever used e-cigarettes; however, this percentage nearly tripled by 2022–2023 to 28% [7]. The dual use of e-cigarettes and conventional cigarettes was most common in the age group of 14–17 years (increased significantly from 0.3% in 2016 to 10.7% in 2023) [16]. Adolescents and young adults (aged 14–30) who have used e-cigarettes are 3.6 times more likely to use conventional cigarettes in the future [17], which potentially can result in an increase in the percentage of adults who smoke conventional combustible cigarettes, threatening a blow to years of efforts by the health agencies to bring down the smoking rates. In addition to the above-mentioned general adverse effects, Castro et al. demonstrated the effects of nicotine exposure on the developing adolescent brain, where nicotine exposure induces long-term changes in cognitive performance and emotional regulation, resulting in increased association with mental health illnesses and great difficulty in quitting smoking if started in early teenage years [18].

Though there is increasing evidence of the potential health risks with the use of e-cigarettes, a recent study states that the adolescent population of Australia does not recognise the harms associated with e-cigarettes and holds positive perceptions about e-cigarette use [19]. Cao et al. stated that discussing e-cigarette use was paramount in diagnosing respiratory injury related to e-cigarette use [9]. However, it is unclear who should be the first point of contact to discuss this pertaining issue. As parents are the primary guardians of children, they could potentially be the first adults to have this discussion; however, cross-sectional studies have reported a lack of parental knowledge and attitude regarding e-cigarettes [20], thus diminishing the likelihood of this discussion. As primary care providers, GPs are uniquely positioned to have these conversations through their trusting and, potentially, long-term relationships with patients and their families. Also, in July 2024, the Australian government issued a new law banning the sale of vaping products to people less than 18 years old without a prescription [21], and thus, the GPs now need to be more prepared to have these conversations with adolescents. However, studies exploring the Knowledge, Attitudes, and Practice (KAP) of GPs have found ambivalence about their knowledge and ability to advise on using e-cigarettes [22,23,24]. There is no recent data on the KAP of GPs practising in Australia regarding e-cigarettes. Therefore, we decided to undertake this pilot study with the aim for it to be the first step to fill in the gaps of information regarding the KAP of the GPs in the state of New South Wales (NSW) in Australia regarding the use of e-cigarettes/vaping in children and adolescents.

## 2. Materials and Methods

The aims of this study were (a) to explore the KAP of GPs around e-cigarette use in children and adolescents and (b) to identify the barriers for GPs to managing children and adolescents using e-cigarettes.

### 2.1. Study Design

The study design was a cross-sectional survey, where the researchers contacted a convenience sample of GPs working in NSW by publicly available phone numbers and email addresses. The local health districts (LHD) contacted were both from metropolitan (Nepean Blue Mountain, Northern Sydney, South Eastern Sydney, South Western Sydney, Sydney, and Western Sydney) and regional and rural areas (Central Coast, Hunter New England, Illawarra Shoalhaven, Mid North Coast, Murrumbidgee, Northern NSW, Southern NSW, and Western NSW). More than 200 GP practices were contacted over three months, almost half of them first by a phone call followed by the invitation email, which included the participant information (see Appendix A). Two inclusion criteria were applied: (1) registered as a GP or GP trainee with AHPRA (Australian Health Practitioner Regulation Authority) and (2) providing care to children and adolescents.

### 2.2. Study Tool

The study tool was a self-administered electronic questionnaire where data were gathered using an online survey. The questionnaire was developed after an extensive review of the literature on the use of e-cigarettes among children and adolescents and the methodology to prepare a questionnaire. There were 33 questions, with the answers designed using the Likert scale responses and five open-ended free-text questions. The questionnaire was piloted with five GPs working in Sydney, and their feedback was considered in the revision. In addition, a link to the free e-learning program KidsQuit [25], providing training on the management of adolescent vaping, was provided at the end of the questionnaire.

### 2.3. Data Collection

The questionnaire was provided as an online link via the Quality Audit Reporting System (QARS) developed by the Clinical Excellence Commission (CEC) of New South Wales (NSW) [26]. Clicking on the online survey link implied participant consent. The study details and link to the survey were also uploaded to the noticeboard of the official website of the Royal Australian College of General Practitioners (RACGP). GP practices in NSW were contacted via publicly available phone numbers, and the survey was sent with an invitation email from December 2022 to February 2023. The researchers conducted visits to GP practices in Western Sydney, and printouts of QR codes were distributed. No identifiable data were collected. Survey reminders were emailed fortnightly twice, followed by monthly for the next two months.

### 2.4. Statistical Analysis

Data analysis was performed using Excel [27] and SPSS [28] statistical packages. For categorical and ordinal-level data, descriptive analysis using absolute (*n*) and relative (%) frequencies was used.

## 3. Results

### 3.1. Demographic Information

The survey recorded responses from 53 GPs (Table 1). The cohort represented well-experienced GPs, where most (81%) were over 40 years old, and 55% had more than 20 years of experience. All the GPs accepted children and adolescents as their patients, where most of them (62%) reviewed less than 20 children and adolescents aged 8–18 years per week. An equal number of GPs had primary qualifications from Australia and overseas (49% each). The socioeconomic indices for areas (SEIFA within Australia) of the practice areas of the GPs ranged from 5 to 97, with the majority of the area with a SEIFA above the 50th percentile. Though multiple GP practices from all over NSW were contacted, much of the response was from the greater Sydney area, with only one response from a regional center (see the map in Appendix A).

### 3.2. Knowledge and Attitude

As a response to the question regarding the use of an e-cigarette by adolescents, about half (45%) of the GPs believed that 20–50% of 12–17-year-olds in Australia have tried an e-cigarette (Figure 1), a number higher than the figure of 12% reported by Guerin et al. [6]. In response to the questions regarding the knowledge about the constituents of e-cigarettes, almost all of them (94%) believed that e-cigarettes contained nicotine either always or sometimes (Figure 2). However, the response was not so unanimous in response to the additional constituents such as propylene glycol, vegetable glycerine, THC, and artificial flavourings, where half of the participants (45%) believed that e-cigarettes contained all of them, and the rest chose one of them to be the only additional constituent (Figure 3).

The beliefs of the participating GPs regarding the potential adverse effects of e-cigarettes, their potential addiction, and their use as a smoking cessation tool are summarised in Table 2. There was strong agreement amongst the GPs that using e-cigarettes was harmful to health, could cause respiratory distress, and could be addictive. Some agreed that using e-cigarettes could cause cardiac disease, burns, and injuries, and non-smokers who used e-cigarettes were more likely to initiate smoking. As a whole, GPs disagreed that e-cigarettes were less addictive than conventional cigarettes and should be recommended as smoking cessation tools, and smokers who did not wish to quit smoking should be encouraged to use e-cigarettes. There was no explicit agreement or disagreement that e-cigarettes were less harmful than conventional cigarettes.

The strongest agreement among the GPs was to the statement that “the use of e-cigarettes can cause respiratory distress”, where 98% of the GPs were either in agreement or strong agreement. The strongest disagreement was with the statement that “e-cigarettes are less addictive than conventional cigarettes”, where 62% of the GPs either disagreed or strongly disagreed. Interestingly, the question most of the GPs could neither agree nor disagree on was “Smokers who do not want to quit smoking should be offered and encouraged to use e-cigarettes”, reflecting an ambiguity of knowledge in this regard.

### 3.3. The Practice of Participating GPs

Overall, the clinical practice of the participating GPs had wide variations with no unanimous approach. Only a third of GPs (34%) discussed e-cigarettes with a child or adolescent attending their clinics for other health concerns. On the contrary, the discussion around conventional cigarettes was higher, with around 60% of GPs discussing it in their clinics (Figure 4). In response to the question regarding the percentage of the population approaching the GPs to discuss e-cigarettes, about half of the GPs (40%) reported that an adolescent or a parent has never approached them to discuss e-cigarettes. When questioned about the practice of GPs using e-cigarettes as a smoking cessation tool, most GPs (79%) had never recommended e-cigarettes to adolescents who smoked conventional cigarettes, although the rest had either rarely or occasionally made that recommendation.

The confidence levels of the participating GPs to advise a child or adolescent regarding the use of e-cigarettes were significantly lower than that for conventional cigarettes, where 36% of GPs were very confident to advise regarding conventional cigarettes as opposed to only 9% for e-cigarettes (Table 3). More concerningly, a third of the GPs (32%) indicated no confidence in giving advice regarding e-cigarettes.

In regards to the clinical approach to discussing e-cigarettes with a child or an adolescent, over half of the GPs (55%) (Figure 5) did not have a structured strategy. More importantly, around 45% of GPs reported this discussion to be difficult (Figure 6).

### 3.4. Barriers to Managing Children Using E-Cigarettes

In the questionnaire, the barriers that GPs faced while managing children using e-cigarettes was an open-ended question, and some of the overlapping responses were as follows: a lack of awareness in the community regarding the potential adverse effects of e-cigarettes, peer pressure on adolescents for using e-cigarettes making them more likely to ignore the advice provided by the GPs, unwillingness of parents and children to participate in a discussion regarding e-cigarettes, lack of comfort of adolescents to talk about vaping in front of their parents during GP appointments limiting the potential for having an open discussion, and inadequate education of GPs to manage and advise children and/or adolescents using e-cigarettes.

In addition, over half of the GPs (53%) were unaware of the educational resources for e-cigarettes.

## 4. Discussion

### 4.1. Principal Findings

The cross-sectional study of 53 GPs practising in Sydney demonstrated that though there were mixed responses from the participating GPs, there was reasonable awareness of the use of e-cigarettes by children and adolescents, the overall harmful nature of e-cigarettes, respiratory side effects, and their addictive potential. More importantly, there was recognition that non-smokers who use e-cigarettes are more likely to take up conventional cigarettes. However, there was lesser recognition of the contents of e-cigarettes, their side effects on the heart, and the occurrence of burns. In practice, a structured conversation about e-cigarettes with children and adolescents in routine clinical practice is lacking, and more importantly, there is a low level of confidence in GPs in having this conversation. The major themes of barriers to managing children using e-cigarettes were lack of awareness in public, peer pressure on teenagers, making them more likely to ignore the advice provided by the GPs, lack of comfort of adolescents to talk about vaping in front of their parents, and inadequate GP education.

### 4.2. Strengths and Limitations

To our knowledge, this is the first cross-sectional study from Australia aimed at understanding the KAP of GPs regarding the use of e-cigarettes in children and adolescents. In a systematic review published by Selamoglu et al. of the KAP of GPs published in 2022 [22] surrounding the prescription of e-cigarettes for smoking cessation, all the included studies were conducted outside Australia, thus lacking representation of Australian GPs. This study’s results are relevant, sourced from an experienced cohort, reflect the current practices, and are comparable to data published internationally [22].

The study’s limitations include the cross-sectional design and the low number of responses despite advertising on relevant websites, personal visits to more than 50 General Practices, and phone calls to more than 200 medical centres, which could lead to a nonresponse bias. There was only one response from regional NSW, so the cohort is unlikely to represent the state-wide or nationwide GP population. Also, as the questionnaire was not statistically validated, such as using principal components analysis or internal analysis, the presence of measurement biases cannot be excluded. In addition, the questions do not allow insight into whether the GPs were approached by the adolescent or by the parent, and thus, it does not provide an idea of the number of adolescents approaching the GPs. Finally, the study was conducted over a short timeframe of 3 months, and perhaps more time would have allowed more responses.

### 4.3. Comparison with Existing Literature

The systematic review published by Selamoglu et al. [22], which included studies from the UK, USA, China, Greece, Belgium, Poland, Slovenia, and Iran, reported that the majority of the GPs believed e-cigarettes to be harmful, though they thought that e-cigarettes were less harmful than regular cigarettes and other tobacco products. These results are comparable to our study, where 96% of GPs believed that e-cigarette use was harmful; however, 39% thought that they were less harmful than conventional cigarettes, and 32% had a neutral opinion. Furthermore, the majority of the participants in the studies reviewed by Selamoglu et al. did not recommend e-cigarettes as a smoking cessation tool, similar to the beliefs of GPs from this study, where 59% disagreed with the statement that e-cigarettes should be recommended as a smoking cessation tool. In addition, Selamoglu et al. reported that under half of the GPs (48.2%) felt confident in their knowledge and capability to respond to patient questions. In our study, this number is even less encouraging as only around one-third of the GPs (34%) were either very or moderately confident in advising children and adolescents regarding the use of e-cigarettes. Our study also demonstrates that the GPs had more confidence in advising about conventional cigarettes than e-cigarettes. This confidence can stem from years of experience and magnitude of education regarding the use of conventional cigarettes, as opposed to the relatively recent development of e-cigarette use and recognition of their potential adverse effects.

It is challenging to compare the results of our study to most other studies [23,24] as participants include physicians from different specialties with a focus on the adult patient population. However, the similarities in the results include physicians not routinely assessing e-cigarette use in their practice [23] and the lack of confidence in advising regarding e-cigarette use among physicians and nurses in the UK [24,29].

Concerningly, studies in the UK, such as Mughal et al. [29], have reported that GPs had lower harm perception, gave less cessation advice, made fewer referrals for e-cigarettes and other smokeless tobacco products, and a third of GPs support using e-cigarettes for smoking cessation. As the population of our study is different from the one in Mughal et al., where the focus was the entire population in the latter, the advice for the use of e-cigarettes as a smoking cessation tool cannot be compared. Nevertheless, our study found that only 8% of GPs have occasionally recommended e-cigarettes for smoking cessation. It is also interesting to note that in 2018, e-cigarettes were advocated for smoking cessation in England due to a perceived notion of them being a harm reduction aid [24]. However, it should be noted that at that time, there were no studies available that demonstrated the adverse effects of using e-cigarettes. Our study reports that in the current era, the perception is clearly the opposite, as 96% of GPs believed that e-cigarettes are harmful to health.

### 4.4. Implications for Clinicians and Policymakers

The paucity of regular discussion, lack of confidence, and significant barriers suggested by this study could potentially be addressed by improving educational opportunities for GPs and the public, as also advised by previous studies, where Stepney et al. [24] reported that practitioners wanted advice from healthcare regulators and Kollath-Cattano et al. [23] stated that physicians expressed an interest in having enhanced discussions about e-cigarettes with their patients and in using patient education material to accomplish this.

It is clear that GPs do recognise that e-cigarettes are not harmless devices, but the complete spectrum of the severity and magnitude of the potential harm is not entirely understood. This half-baked knowledge is possibly the cause of low confidence levels, which can impact the services provided to children and adolescents. Our pilot study can provide the initial information to the policymakers in Australia regarding the practices of GPs and the potential barriers in providing the services to the children and adolescents of the community and can start a conversation into collecting large-scale data, improving educational resources to the GPs, and providing confidence-building measures.

### 4.5. Unanswered Questions and Future Research

Although this study was aimed to explore the KAP of GPs practising all over the state of NSW, the majority of the responses were only from the greater Sydney area. As the cohort does not represent the GPs practising in rural areas of NSW, there is no information regarding the knowledge and practices of GPs in the regional towns. We recommend that future research include these demographics to understand the true KAP of GPs practising all over NSW.

Also, future research can involve participation from adolescents and study their expectations from GPs as well as the barriers they face in seeking advice from a healthcare professional. Also, future research projects can study the training requirements of the GPs in managing children and adolescents regarding the use of e-cigarettes.

## 5. Conclusions

To conclude, the participating GPs from Sydney are somewhat knowledgeable about the potential adverse effects of the use of e-cigarettes in children and adolescents, though there is a lack of knowledge of the complete spectrum of adverse effects and, more importantly, there is a paucity of a structured approach to discussing the use of e-cigarettes with children and adolescents and there is a low level of confidence in addressing these issues. There is a need for educational interventions for GPs to increase awareness of the potential adverse effects of using e-cigarettes and build confidence in providing management to children and adolescents regarding the use of e-cigarettes.

## Figures and Tables

**Figure 1 ijerph-21-01215-f001:**
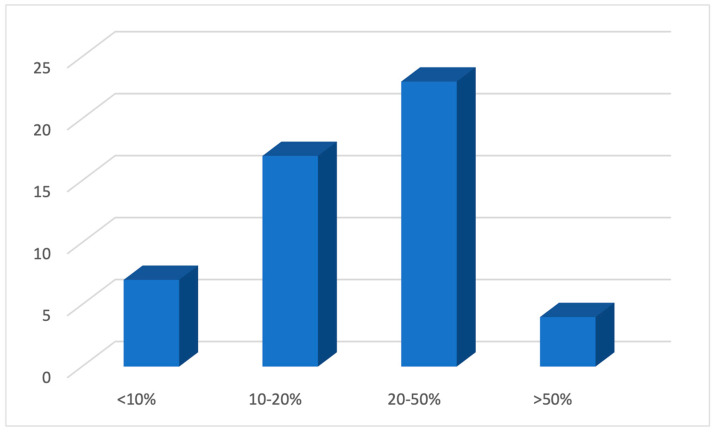
The belief of GPs regarding the percentage of 12 to 17-year-olds in Australia who have tried an e-cigarette.

**Figure 2 ijerph-21-01215-f002:**
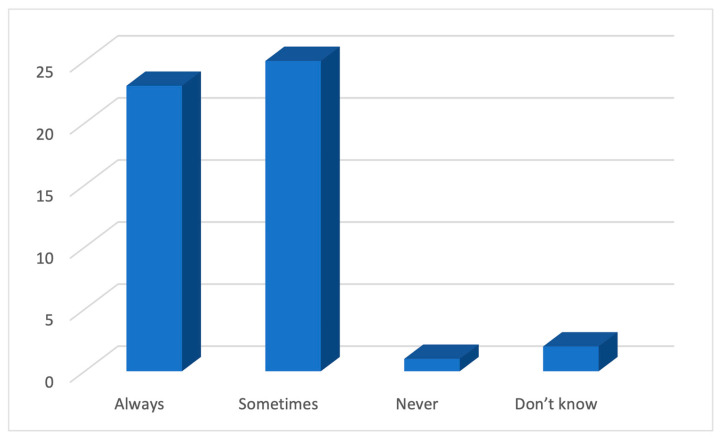
The belief of GPs regarding whether e-cigarettes contain nicotine.

**Figure 3 ijerph-21-01215-f003:**
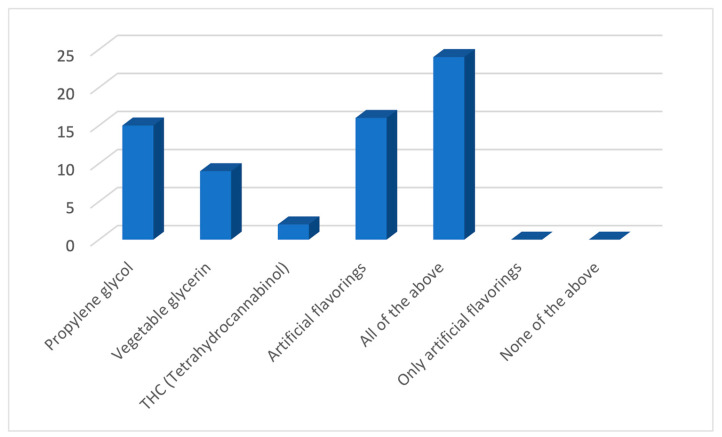
Knowledge of participating GPs regarding the contents of e-cigarettes.

**Figure 4 ijerph-21-01215-f004:**
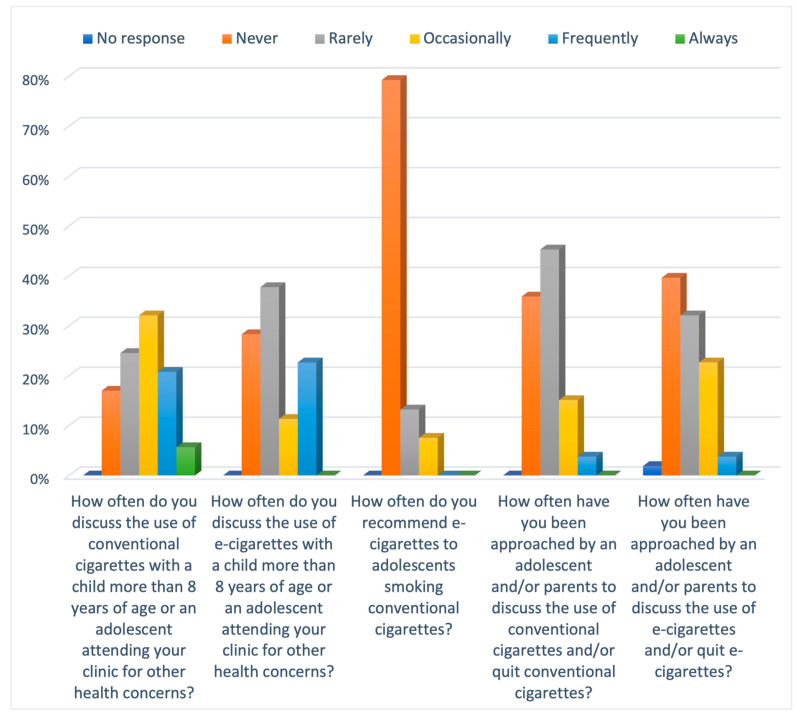
Bar chart illustrating the Likert scale responses to practices of GPs.

**Figure 5 ijerph-21-01215-f005:**
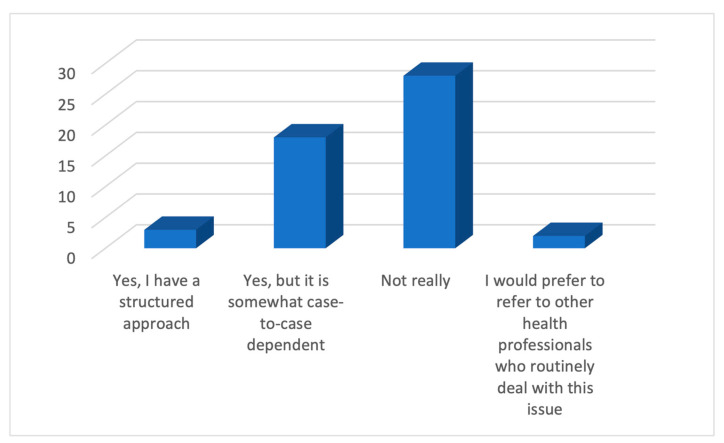
Responses of GPs on whether they have a strategy for discussing e-cigarettes.

**Figure 6 ijerph-21-01215-f006:**
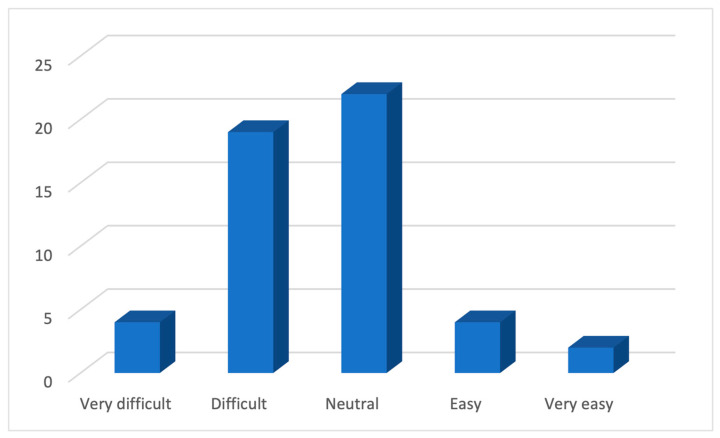
The level of difficulty faced by the GPs in discussing e-cigarettes.

**Table 1 ijerph-21-01215-t001:** Participant characteristics (*n* = 53).

Characteristics of Participating GP	Number (*n*)	Percentage (%)
Age		
- <30-years-old	4	7
- 30–39-years-old	5	9
- 40–49-years-old	16	30
- 50–59-years-old	12	23
- >60-years-old	15	28
- No response	1	2
Gender		
- Male	24	45
- Female	29	55
Years of professional practice:		
- <2 years	3	6
- 2–5 years	8	15
- 5–10 years	3	6
- 10–20 years	10	19
- >20 years	29	55
Estimated number of children and adolescents between 8 and 18 years of age reviewed per week:		
- <20	33	62
- 20–50	19	36
- 50–100	1	2
- >100	0	0
Country of primary qualification:		
- Australia	26	49
- Others	26	49
- No answer	1	2
Socio-Economic Indexes for Areas (SEIFA)		
- <50 percentile within Australia	11	21
- 50–75 percentile within Australia	19	36
- >75 percentile within Australia	21	40
No response	2	3

**Table 2 ijerph-21-01215-t002:** Beliefs of GPs regarding e-cigarettes.

Statement	Strongly Disagree *n* (%)	Disagree *n* (%)	Neither Agree nor Disagree *n* (%)	Agree *n* (%)	Strongly Agree *n* (%)	Mean *
The use of e-cigarettes is harmful to the health of the user	0 (0)	0 (0)	2 (4)	17 (32)	34 (64)	4.6
E-cigarettes are less harmful than conventional cigarettes	6 (11)	15 (28)	17 (32)	14 (26)	1 (2)	2.8
The use of e-cigarettes can cause respiratory distress	0 (0)	0 (0)	1 (2)	24 (45)	28 (53)	4.5
The use of e-cigarettes can cause cardiac disease	0 (0)	2 (4)	5 (9)	27 (51)	19 (36)	4.2
The use of e-cigarettes can potentially cause burns and injuries	0 (0)	1 (2)	7 (13)	29 (55)	16 (30)	4.1
Non-smokers who use e-cigarettes are more likely to initiate cigarette smoking	0 (0)	2 (4)	9 (17)	24 (45)	18 (34)	4.1
Among non-smokers, e-cigarette use results in dependence on e-cigarettes	0 (0)	1 (2)	5 (9)	30 (57)	17 (32)	4.2
E-cigarettes can be addictive	0 (0)	0 (0)	2 (4)	23 (43)	28 (53)	4.5
E-cigarettes are less addictive than conventional cigarettes	9 (17)	24 (45)	13 (26)	6 (11)	1 (2)	2.4
E-cigarettes should be recommended as a smoking cessation tool	10 (19)	21 (40)	16 (30)	3 (6)	3 (6)	2.4
Smokers who do not want to quit smoking should be offered and encouraged to use e-cigarettes	9 (17)	15 (28)	19 (36)	9 (17)	1 (2)	2.6

* 5-point Likert scale range of mean: Strongly disagree: 1–1.8, Disagree: 1.9–2.6, Neither disagree nor agree: 2.7–3.4, Agree: 3.5–4.2, Strongly agree: 4.3–5.0.

**Table 3 ijerph-21-01215-t003:** Confidence of GPs regarding advising about e-cigarettes.

Statement	Very Confident *n* (%)	Moderately Confident *n* (%)	Somewhat Confident *n* (%)	Not Confident at All *n* (%)
How confident do you feel advising a child or an adolescent about the use of conventional cigarettes?	19 (36)	11 (21)	19 (36)	4 (8)
How confident do you feel advising a child or an adolescent about the use of e-cigarettes?	5 (9)	12 (23)	19 (36)	17 (32)

## Data Availability

The datasets used and/or analyzed during the current study are available from the corresponding author upon reasonable request.

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
