# Peer review of "Exploring General Practitioners’ Knowledge, Attitudes, and Practices towards E-Cigarette Use/Vaping in Children and Adolescents: A Pilot Cross-Sectional Study in Sydney"

_ijerph, 2024, doi:10.3390/ijerph21091215_

Round 1

Reviewer 1 Report

Comments and Suggestions for Authors

line 35 - no mention of VG?

line 40 - this study is 2018 which seems quite old. I am sure the new AIHW covers this point, and would be a much better source anyway'.

line 44 - refer to handful of EVALI cases that occurred outside US, including the Aus cases.

line 45 - ref for arterial stiffness should be added.

line 48 - add ref for claim RE link to taking up smoking.

line 51 - not a very relevant line when EVALI was  not really a thing here - just refer to respiratory injury instead.

line 57 - be good to mention the recent change in Aus law that adults can obtain nicotine ecigs at a pharmacy without prescription now but under 18s still require a prescription, so GPs need to be prepared for these discussions.

Study design - geographic and economic status of areas  covered? How many in total were contacted? How many by phone and how many by email? What did you say when contacting by phone? What when contacting by email?

Ethical approval - this work should have been approved by a HREC. Please provide details.

Table 1 - add socioeconomic status (should be available for their LGAs).

Results - prefer to see x axis as % of respondants as this is how paper represent the results.

Table 2 results description text - this section should be re-written to discuss question in terms of largest group between agreed+ st agree and disagree +strong disagree or neutral percentage.

Limitations - add limitation - questions does not allow you to discern if it was child or parent asking, and I think that would have been valuable to.

Make sure figures are large enough that questions can be read as this is only place they are in the paper.

3.2. - What about adolescents appointments often having parents present so less likely to feel comfortable asking question/say they vape in front of parent?

line 187 - add what country this study was done int.

PHE report - should be made clear that this report's finding was not based on any studies actually looking at harm as at the time there were almost no studies done.

Reviewer 2 Report

Comments and Suggestions for Authors

This survey study assessed knowledge, attitudes and practice of GP’s in Sydney, Australia around youth e-cigarette use. The authors do a nice job of collecting rather comprehensive information on attitudes and practice around youth vaping. Given the recruitment approach, this study represents a very small number of pediatric providers in a very specific location thus findings are not generalizable to pediatric providers. The findings of this local group of providers may be highly influenced by local advertising, public health campaigns, local CME opportunities, and local policy/laws. In addition, the study uses questions which are not validated. Overall, these issues greatly limits the utility of the findings to a broader audience. I think this study could be published in a journal with a regional audience to where the study occurred.

Introduction

Line 40: would add date of when Australia vaping rates were obtained

Line 48: Please add citation and context. Is this amongst all e-cigarette users. Is this adult/youth specific

Line 50: There are numerous trials now showing significant impact of e-cigarettes in helping people quit smoking. I would delete this sentence as it appears to be outdated. If this is referring to youth smoking cessation, please clarify.

Line 58: Please spell out NSW

Methods

-please add the dates that the study occurred under section '2.3. Data Collection'. If data is several years old, I worry about it’s relevance as the e-cigarette environment continues to rapidly change.

Results

Line 98: ‘reviewed’ was not a term I was familiar with for patient encounters. This was easily determined after reading the table. The table states ‘reviewed per week’. I recommend adding the ‘per week’ qualifier at the end of the sentence on line 99.

Line 134: change ‘better’ to ‘higher’

Figure 4: This bar chart feels difficult to read and may be more readable in a different format.

Discussion

Line 164: Again, regarding e-cigarettes as a cessation tool. There is evidence of the effectiveness, though this is specific to adults.

Line 167: I do not understand how ‘peer pressure’ on adolescents is a barrier to a GP discussing e-cigarettes with youth. It seems like peer pressure affects youth use of e-cigarettes but shouldn’t impact a GP’s intervention.

Comments on the Quality of English Language

No major concerns.

Reviewer 3 Report

Comments and Suggestions for Authors

The purpose of this article was to 1) explore the knowledge, attitudes, and practices of general practitioners (GP) around e-cigarette (e-cig) use in children and adolescents in Sydney, Australia, and 2) to identify barriers for GPs to manage children and adolescents using e-cigs. Overall, this topic is important for the field of public health. Improvements to the paper are listed below for each section of the manuscript.

Introduction

Page 1, Lines 39-40: The authors provide the prevalence of ever-use or those who have tried an e-cig among 12–17-year-olds in Australia; however, there is no mention of the current use of e-cigs, especially those who use them on 20 or more days during the month. It may be helpful for readers to understand the impact of this topic if they know the current use of e-cigs and dual use of e-cigs and combustible cigarettes. The authors do not mention the effects of nicotine on the developing adolescent brain in the Introduction section.

Page 2, lines 47-50: The authors state that "Those using e-cigarettes are three times more likely to smoke combustible tobacco," yet one of the articles cited indicates that combustible tobacco smoking among youth globally and in Australia has significantly dropped. Please clarify this discrepancy. If e-cigs leads youth to smoke combustible tobacco, then how are they three times more likely to smoke tobacco if the data does not show this?

Page 2, Line 58: What is NSW? Suggest adding what this acronym is when spelled out.

Page 2, Line 59: Are there any articles on the inclusion of e-cigarettes in practitioner training? Suggest including a review of the literature on the training of practitioners, as this may provide some background on why or why not, general practitioners are not providing sufficient consultation to youth about the harms of e-cig use and the treatments that may be available.

Methods

Page 3, Lines 91-94: How was missing data handled? There were no tests of significance calculated between variables. This seemed to be a mostly descriptive study.

Page 3, Table 1: Under the “Country of primary qualification:” no list or note was provided on the definition of “Others.” It would have been nice to see the names of the other countries or the number of other countries where practitioners received their primary qualification.

Discussion

Page 8, Lines 210-211: As mentioned in suggestions for the Introduction, there was no literature that mentioned the training practitioners received for discussing use of e-cigs with youth. It would be beneficial to state the literature on this topic at the beginning, so when it is mentioned in the Discussion a more robust synthesis could be provided.

Round 2

Reviewer 3 Report

Comments and Suggestions for Authors

The co-authors have responded to this reviewer's comments to satisfaction.

Comments on the Quality of English Language

Minor edits to the quality of English language are needed. Some areas are missing "s" when referring the age of the population in the introduction. Articles such as "the" are missing throughout the paper.

Author Response

We thank the reviewer for the comments. We have updated the whole article and edited accordingly.